# Laplacian Networks: Bounding Indicator Function Smoothness for Neural Networks Robustness

## Abstract

For the past few years, Deep Neural Network (DNN) robustness has become a question of paramount importance. As a matter of fact, in sensitive settings misclassification can lead to dramatic consequences. Such misclassifications are likely to occur when facing adversarial attacks, hardware failures or limitations, and imperfect signal acquisition. To address this question, authors have proposed different approaches aiming at increasing the robustness of DNNs, such as adding regularizers or training using noisy examples. In this paper we propose a new regularizer built upon the Laplacian of similarity graphs obtained from the representation of training data at each layer of the DNN architecture. This regularizer penalizes large changes (across consecutive layers in the architecture) in the distance between examples of different classes, and as such enforces smooth variations of the class boundaries. Since it is agnostic to the type of deformations that are expected when predicting with the DNN, the proposed regularizer can be combined with existing ad-hoc methods. We provide theoretical justification for this regularizer and demonstrate its effectiveness to improve robustness of DNNs on classical supervised learning vision datasets.

## 1 Introduction

Deep Neural Networks (DNNs) provide state-of-the-art performance in many challenges in machine learning (He et al., 2016; Wu et al., 2016). Their ability to achieve good generalization is often explained by the fact they use very few priors about data (LeCun et al., 2015). On the other hand, their strong dependency on data may lead to focus on biased features of the training dataset, resulting in a nonrobust classification performance.

In the literature, authors have been interested in studying the robustness of DNNs in various conditions. These conditions include:

- Robustness to isotropic noise, i.e., small isotropic variations of the input (Mallat, 2016), typically meaning that the network function leads to a small Lipschitz constant.
- Robustness to adversarial attacks, which can exploit knowledge about the network parameters or the training dataset (Szegedy et al., 2013; Goodfellow et al., 2014).
- Robustness to implementation defects, which can result in only approximately correct computations (Hubara et al., 2017).

To improve DNN robustness, three main families of solutions have been proposed in the literature. The first one involves enforcing smoothness, as measured by a Lipschitz constant, in the operators and having a minimum separation margin (Mallat, 2016). A similar approach has been proposed in (Cisse et al., 2017), where the authors restrict the function of the network to be contractive. A second class of methods use intermediate representations obtained at various layers to perform the prediction (Papernot and McDaniel, 2018). Finally, in (Kurakin et al., 2016; Pezeshki et al., 2016; Madry et al., 2018), the authors propose to

train the network using noisy inputs so that it better generalizes to this type of noise. This has been shown to improve the robustness of the network to the specific type of noise used during training, but it is not guaranteed that this robustness would be extended to other types of deformations.

In this work, we introduce a new regularizer that does not focus on a specific type of deformation, but aims at increasing robustness in general. As such, the proposed regularizer can be combined with other existing methods. It is inspired by recent developments in Graph Signal Processing (GSP) (Shuman et al., 2013). GSP is a mathematical framework that extends classical Fourier analysis to complex topologies described by graphs, by introducing notions of frequency for signals defined on graphs. Thus, signals that are smooth on the graph (i.e., change slowly from one node to its neighbors) will have most of their energy concentrated in the low frequencies.

The proposed regularizer is based on constructing a series of graphs, one for each layer of the DNN architecture, where each graph captures the similarity between all training examples given their intermediate representation at that layer. Our proposed regularizer penalizes large changes in the smoothness of class indicator vectors (viewed here as graph signals) from one layer to the next. As a consequence, the distances between pairs of examples in different classes are only allowed to change slowly from one layer to the next. Note that because we use deep architectures, the regularizer does not prevent the smoothness from achieving its maximum value, but constraining the size of changes from layer to layer increases the robustness of the network function by controlling the distance to the boundary region, as supported by experiments in Section 4.

The outline of the paper is as follows. In Section 2 we present related work. In Section 3 we introduce the proposed regularizer. In Section 4 we evaluate the performance of our proposed method in various conditions and on vision benchmarks. Section 5 summarizes our conclusions.

## 2 Related work

DNN robustness may refer to many different problems. In this work we are mostly interested in the stability to deformations (Mallat, 2016), or noise, which can be due to multiple factors mentioned in the introduction. The most studied stability to deformations is in the context of adversarial attacks. It has been shown that very small imperceptible changes on the input of a trained DNN can result in missclassification of the input (Szegedy et al., 2013; Goodfellow et al., 2014). These works have been primordial to show that DNNs may not be as robust to deformations as the test accuracy benchmarks would have lead one to believe. Other works, such as (Recht et al., 2018), have shown that DNNs may also suffer from drops in performance when facing deformations that are not originated from adversarial attacks, but simply by re-sampling the test images.

Multiple ways to improve robustness have been proposed in the literature. They range from the use of a model ensemble composed of $k$-nearest neighbors classifiers for each layer (Papernot and McDaniel, 2018), to the use of distillation as a mean to protect the network (Papernot et al., 2016a). Other methods introduce regularizers (Gu and Rigazio, 2014), control the Lipschitz constant of the network function (Cisse et al., 2017) or implement multiple strategies revolving around using deformations as a data augmentation procedure during the training phase (Goodfellow et al., 2014; Kurakin et al., 2016; Moosavi Dezfooli et al., 2016).

Compared to these works, our proposed method can be viewed as a regularizer that penalizes large deformations of the class boundaries throughout the network architecture, instead of focusing on a specific deformation of the input. As such, it can be combined with other mentioned strategies. Indeed, we demonstrate that the proposed method can be implemented in combination with (Cisse et al., 2017), resulting in a network function such that small variations to the input lead to small variations in the decision, as in (Cisse et al., 2017), while limiting the amount of change to the class boundaries. Note that our approach does

not require using training data affected by a specific deformation, and our results could be further improved if such data were available for training, as shown in the Appendix.

As for combining GSP and machine learning, this area has sparked interest recently. For example, the authors of (Gripon et al., 2018) show that it is possible to detect overfitting by tracking the evolution of the smoothness of a graph containing only training set examples. Another example is in (Anirudh et al., 2017) where the authors introduce different quantities related to GSP that can be used to extract interpretable results from DNNs. In (Svoboda et al., 2018) the authors exploit graph convolutional layers (Bronstein et al., 2017) to increase the robustness of the network.

To the best of our knowledge, this is the first use of graph signal smoothness as a regularizer for deep neural network design.

## 3 Methodology

### 3.1 Similarity preset and postset graphs

Consider a deep neural network architecture. Such a network is obtained by assembling layers of various types. Of particular interest are layers of the form $\mathbf{x}^\ell \mapsto \mathbf{x}^{\ell+1} = h^\ell(\mathbf{W}^\ell \mathbf{x}^\ell + \mathbf{b}^\ell)$, where $h^\ell$ is a nonlinear function, typically a ReLU, $\mathbf{W}^\ell$ is the weight tensor at layer $\ell$, $\mathbf{x}^\ell$ is the intermediate representation of the input at layer $\ell$ and $\mathbf{b}^\ell$ is the corresponding bias tensor. Note that strides or pooling may be used. Assembling can be achieved in various ways: composition, concatenation, sums... so that we obtain a global function $f$ that associates an input tensor $\mathbf{x}^0$ to an output tensor $\mathbf{y} = f(\mathbf{x}^0)$.

When computing the output $\mathbf{y}$ associated with the input $\mathbf{x}^0$, each layer $\ell$ of the architecture processes some input $\mathbf{x}^\ell$ and computes the corresponding output $\mathbf{y}^\ell = h^\ell(\mathbf{W}^\ell \mathbf{x}^\ell + \mathbf{b}^\ell)$. For a given layer $\ell$ and a batch of $b$ inputs $\mathcal{X} = \{\mathbf{x}_1, \ldots, \mathbf{x}_b\}$, we can obtain two sets $\mathcal{X}^\ell = \{\mathbf{x}_1^\ell, \ldots, \mathbf{x}_b^\ell\}$, called the *preset*, and $\mathcal{Y}^\ell = \{\mathbf{y}_1^\ell, \ldots, \mathbf{y}_b^\ell\}$, called the *postset*.

Given a similarity measure $s$ on tensors, from a preset we can build the similarity preset matrix: $\mathbf{M}_{pre}^\ell[i,j] = s(\mathbf{x}_i^\ell, \mathbf{x}_j^\ell), \forall 1 \leq i, j \leq b$, where $\mathbf{M}[i,j]$ denotes the element at line $i$ and column $j$ in $\mathbf{M}$. The postset matrix is defined similarly.

Consider a similarity (either preset or postset) matrix $\mathbf{M}^\ell$. This matrix can be used to build a $k$-nearest neighbor similarity weighted graph $G^\ell = \langle V, \mathbf{A}^\ell \rangle$, where $V = \{1, \ldots, b\}$ is the set of vertices and $\mathbf{A}^\ell$ is the weighted adjacency matrix defined as:

$$\mathbf{A}^\ell[i,j] = \begin{cases} \mathbf{M}^\ell[i,j] & \text{if } \mathbf{M}^\ell[i,j] \in \arg\max_{i' \neq j} (\mathbf{M}^\ell[i',j], k) \\ & \bigcup \arg\max_{j' \neq i} (\mathbf{M}^\ell[i,j'], k) \quad , \forall i, j \in V, \\ 0 & \text{otherwise} \end{cases} \tag{1}$$

where $\arg\max_i(a_i, k)$ denotes the indices of the $k$ largest elements in $\{a_1, \ldots, a_b\}$. Note that by construction $\mathbf{A}^\ell$ is symmetric.

### 3.2 Smoothness of label signals

Given a weighted graph $G^\ell = \langle V, \mathbf{A}^\ell \rangle$, we call Laplacian of $G^\ell$ the matrix $\mathbf{L}^\ell = \mathbf{D}^\ell - \mathbf{A}^\ell$, where $\mathbf{D}^\ell$ is the diagonal matrix such that: $\mathbf{D}^\ell[i,i] = \sum_j \mathbf{A}^\ell[i,j], \forall i \in V$. Because $\mathbf{L}^\ell$ is symmetric and real-valued, it can be written:

$$\mathbf{L}^\ell = \mathbf{F}^\ell \mathbf{\Lambda}^\ell \mathbf{F}^{\ell \top}, \tag{2}$$

where $\mathbf{F}$ is orthonormal and contains eigenvectors of $\mathbf{L}^\ell$ as columns, $\mathbf{F}^\top$ denotes the transpose of $\mathbf{F}$, and $\mathbf{\Lambda}$ is diagonal and contains eigenvalues of $\mathbf{L}^\ell$ is ascending order. Note that the constant vector $\mathbf{1} \in \mathbb{R}^b$ is an eigenvector of $\mathbf{L}^\ell$ corresponding to eigenvalue 0. Moreover, all eigenvalues of $\mathbf{L}^\ell$ are nonnegative. Consequently, $\mathbf{1}/\sqrt{n}$ can be chosen as the first column in $\mathbf{F}$.

Consider a vector $\mathbf{s} \in \mathbb{R}^b$, we define $\hat{\mathbf{s}}$ the Graph Fourier Transform (GFT) of $\mathbf{s}$ on $G^\ell$ as (Shuman et al., 2013):

$$\hat{\mathbf{s}} = \mathbf{F}^\top \mathbf{s}. \tag{3}$$

Because the order of the eigenvectors is chosen so that the corresponding eigenvalues are in ascending order, if only the first few entries of $\hat{\mathbf{s}}$ are nonzero that indicates that $\mathbf{s}$ is low frequency (smooth). In the extreme case where only the first entry of $\hat{\mathbf{s}}$ is nonzero we have that $\mathbf{s}$ is constant (maximum smoothness). More generally, smoothness $\sigma^\ell(\mathbf{s})$ of a signal $\mathbf{s}$ can be measured using the quadratic form of the Laplacian:

$$\sigma^\ell(\mathbf{s}) = \mathbf{s}^\top \mathbf{L}^\ell \mathbf{s} = \sum_{i,j=1}^b \mathbf{A}^\ell[i,j](\mathbf{s}[i] - \mathbf{s}[j])^2 = \sum_{i=1}^b \mathbf{\Lambda}^\ell[i,i]\hat{\mathbf{s}}[i]^2, \tag{4}$$

where we note that $\mathbf{s}$ is smoother when $\sigma^\ell(\mathbf{s})$ is smaller.

In this paper we are particularly interested in smoothness of the label signals. We call *label signal* $\mathbf{s}_c$ associated with class $c$ a binary ($\{0,1\}$) vector whose nonzero coordinates are the ones corresponding to input vectors of class $c$. In other words, $\mathbf{s}_c[i] = 1 \Leftrightarrow (\mathbf{x}_i$ is in class $c), \forall 1 \leq i \leq b$. Using Equation (4), we obtain that the smoothness of the label signal $\mathbf{s}_c$ is the sum of similarities between examples in distinct classes. Thus a smoothness of 0 means that examples in distinct classes have 0 similarity.

Denote $u$ the last layer of the architecture: $\mathbf{y}_i^u = \mathbf{y}_i, \forall i$. Note that in typical settings, where outputs of the networks are one-hot-bit encoded and no regularizer is used, at the end of the learning process it is expected that $\mathbf{y}_i^\top \mathbf{y}_j \approx 1$ if $i$ and $j$ belong to the same class, and $\mathbf{y}_i^\top \mathbf{y}_j \approx 0$ otherwise.

Thus, assuming that cosine similarity is used to build the graph, the last layer smoothness for all $c$ would be $\sigma_{post}^u(\mathbf{s}_c) \approx 0$, since edge weights between nodes having different labels will be close to zero given Equation (4). More generally, smoothness of $\mathbf{s}_c$ at the preset or postset of a given layer measures the average similarity between examples in class $c$ and examples in other classes ($\sigma(\mathbf{s}_c)$ decreases as the weights of edges connecting nodes in different classes decrease). Because the last layer can achieve $\sigma(\mathbf{s}_c) \approx 0$, we expect the smoothness metric $\sigma$ at each layer to decrease as we go deeper in the network. Next we introduce a regularization strategy that limits how much $\sigma$ can decrease from one layer to the next and can even prevent the last layer from achieving $\sigma(\mathbf{s}_c) = 0$. This will be shown to improve generalization and robustness. The theoretical motivation for this choice is discussed in Section 3.4.

### 3.3 Proposed regularizer

#### 3.3.1 Definition

We propose to measure the deformation induced by a given layer $\ell$ in the relative positions of examples by computing the difference between label signal smoothness before and after the layer, averaged over all labels:

$$\delta_\sigma^\ell = \left| \sum_c \left[ \sigma_{post}^\ell(\mathbf{s}_c) - \sigma_{pre}^\ell(\mathbf{s}_c) \right] \right|. \tag{5}$$

These quantities are used to regularize modifications made to each of the layers during the learning process.

*Remark 1:* Since we only consider label signals, we solely depend on the similarities between examples that belong to distinct classes. In other words, the regularizer only focuses on the boundary region, and does not vary if the distance between examples of the same label grows or shrinks. This is because forcing similarities between examples of a same class to evolve slowly could prevent the network to train appropriately.

*Remark 2:* Compared with (Cisse et al., 2017), there are three key differences that characterize the proposed regularizer:

1. Not all pairwise distances are taken into account in the regularization; only distances between examples corresponding to different classes play a role in the regularization.

2. We allow a limited amount of both contraction and dilation of the metric space. Experimental work (e.g. (Gripon et al., 2018; Papernot and McDaniel, 2018)) has

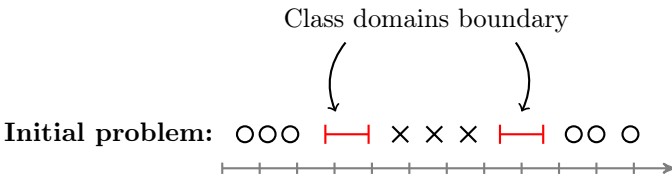

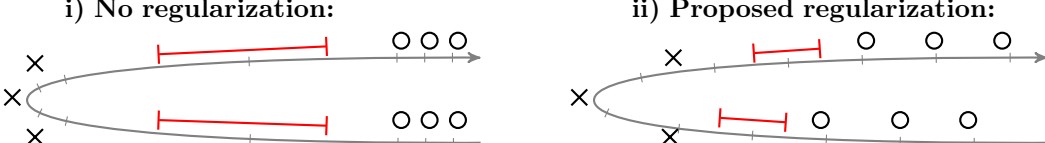

Figure 1: Illustration of the effect of our proposed regularizer. In this example, the goal is to classify circles and crosses (top). Without use of regularizers (bottom left), the resulting embedding may considerably stretch the boundary regions (as illustrated by the irregular spacing between the tics). Forcing small variations of smoothness of label signals (bottom right), we ensure the topology is not dramatically changed in the boundary regions.

    shown that the evolution of metric spaces across DNN layers is complex, and thus restricting ourselves to contractions only could lead to lower overall performance.

3. The proposed criterion is an average (sum) over all distances, rather than a stricter criterion (e.g. Lipschitz), which would force each pair of vectors $(\mathbf{x}_i, \mathbf{x}_j)$ to obey the constraint.

**Illustrative example:**

In Figure 1 we depict a toy illustrative example to motivate the proposed regularizer. We consider here a one-dimensional two-class problem. To linearly separate circles and crosses, it is necessary to group all circles. Without regularization (setting i)), the resulting embedding is likely to increase considerably the distance between examples and the size of the boundary region between classes. In contrast, by penalizing large variations of the smoothness of label signals (setting ii)), the average distance between circles and crosses must be preserved in the embedding domain, resulting in a more precise control of distances within the boundary region.

### 3.4 Motivation: label signal bandwidth and powers of the Laplacian

Recent work (Anis et al., 2017) develops an asymptotic analysis of the bandwidth of label signals, $BW(\mathbf{s})$, where bandwidth is defined as the highest non-zero graph frequency of $\mathbf{s}$, i.e., the nonzero entry of $\hat{\mathbf{s}}$ with the highest index. An estimate of the bandwidth can be obtained by computing:

$$BW_m(\mathbf{s}) = \left( \frac{\mathbf{s}^\top \mathbf{L}^m \mathbf{s}}{\mathbf{s}^\top \mathbf{s}} \right)^{(1/m)} \tag{6}$$

for large $m$. This can be viewed as a generalization of the smoothness metric of (4). (Anis et al., 2017) shows that, as the number of labeled points $\mathbf{x}$ (assumed drawn from a distribution $p(\mathbf{x})$) grows asymptotically, the bandwidth of the label signal converges in probability to the supremum of $p(\mathbf{x})$ in the region of overlap between classes. This motivates our work in three ways.

First, it provides theoretical justification to use $\sigma^\ell(\mathbf{s})$ for regularization, since lower values of $\sigma^\ell(\mathbf{s})$ are indicative of better separation between classes. Second, the asymptotic analysis suggests that using higher powers of the Laplacian would lead to better regularization, since estimating bandwidth using $BW_m(\mathbf{s})$ becomes increasingly accurate as $m$ increases. Finally, this regularization can be seen to be protective against specializing by preventing $\sigma^\ell(\mathbf{s})$

Middle layer                                    Deep layer

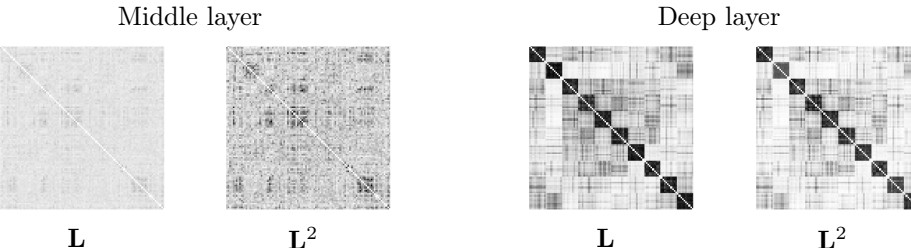

**L**                     $\mathbf{L}^2$                    **L**                     $\mathbf{L}^2$

Figure 2: Sample of a Laplacian and squared Laplacian of similarity graphs in a trained vanilla architecture. Examples of the batch have been ordered so that those belonging to a same class are consecutive. Dark values correspond to high similarity.

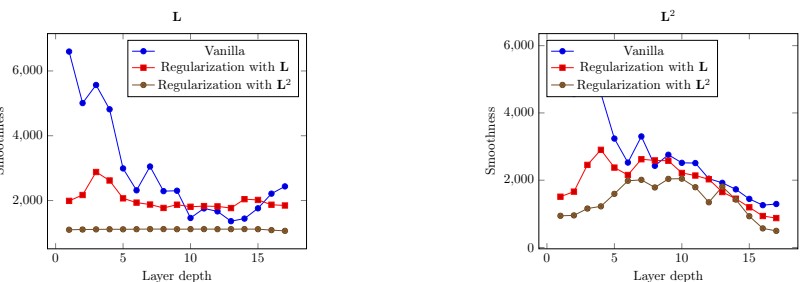

Figure 3: Evolution of smoothness of label signals as a function of layer depth, and for various regularizers and choice of $m$, the power of the Laplacian matrix.

from decreasing "too fast". For most problems of interest, given a sufficiently large amount of labeled data available, it would be reasonable to expect the bandwidth of **s** not to be arbitrarily small, because the classes cannot be exactly separated, and thus a network that reduces the bandwidth too much can result in being biased by the training set.

### 3.5 Analysis of the Laplacian powers

In Figure 2 we depict the Laplacian and squared Laplacian of similarity graphs obtained at different layers in a trained vanilla architecture. On the deep layers, we can clearly see blocks corresponding to the classes, while the situation in the middle layer is not as clear. This figure illustrates how using the squared Laplacian helps modifying the distances to improve separation. Note that we normalize the squared Laplacian values by dividing them by the highest absolute value.

In Figure 3, we plot the average evolution of smoothness of label signals over 100 batches, as a function of layer depth in the architecture, and for different choices of the regularizer. In the left part, we look at smoothness measures using the Laplacian. In the right part, we use the squared Laplacian. We can clearly see the effectiveness of the regularizer in enforcing small variations of smoothness across the architecture. Note that for model regularized with $\mathbf{L}^2$, changes in smoothness measured by $\mathbf{L}$ are not easy to see. This seems to suggest that some of the gains achieved via $\mathbf{L}^2$ regularization come in making changes that would be "invisible" when looking at the layers from the perspective of $\mathbf{L}$ smoothness. The same normalization from Figure 2 is used for $\mathbf{L}^2$.

## 4 Experiments

In the following paragraphs we evaluate the proposed method using various tests. We use the well known CIFAR-10 (Krizhevsky and Hinton, 2009) dataset made of tiny images. As far as the DNN is concerned, we use the same PreActResNet (He et al., 2016) architecture for all tests, with 18 layers. All inputs, including those on the test set, are normalized based on the mean and standard deviation of the images of the *training* set. In all figures, P are

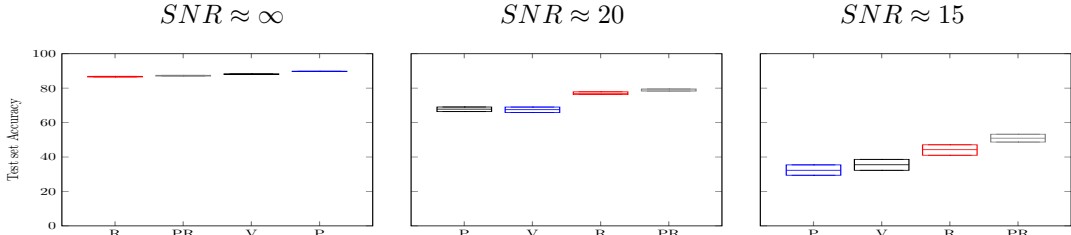

Figure 4: Test set accuracy under Gaussian noise with varying signal-to-noise ratio.

Parseval trained networks, R are networks trained with the proposed regularizer and V are vanilla networks. More details and experiments can be found at the Appendix.

We depict the obtained results using box plots where data is aggregated from 10 different networks corresponding to different random seeds and batch orders. In the first experiment (left most plot) in Figure 4, we plot the baseline accuracy of the models on the clean test set (no deformation is added at this point). These experiments agree with the claim from (Cisse et al., 2017) where the authors show that they are able to increase the performance of the network on the clean test set. We observe that our proposed method leads to a minor decrease of performance on this test. However, we see in the following experiments that this is mitigated with increased robustness to deformations. Such a trade-off between robustness and accuracy has already been discussed in the literature (Fawzi et al., 2018).

### 4.1 Isotropic deformation

In this scenario we evaluate the robustness of the network function to small isotropic variations of the input. We generate 40 different deformations using random variables $\mathcal{N}(0, 0.25)$ which are added to the test set inputs. Note that they are scaled so that $SNR \approx 15$ and $SNR \approx 20$. The middle and right-most plots from Figure 4 show that the proposed method increases the robustness of the network to isotropic deformations. Note that in both scenarios the best results are achieved by combining Parseval training and our proposed method (lower-most box on both figures).

### 4.2 Adversarial Robustness

We next evaluate robustness to adversarial inputs, which are specifically built to fool the network function. Such adversarial inputs can be generated and evaluated in multiple ways. Here we implement two approaches: first a mean case of adversarial noise, where the adversary can only use one forward and one backward pass to generate the deformations, and second a worst case scenario, where the adversary can use multiple forward and backward passes to try to find the smallest deformation that will fool the network.

For the first approach, we add the scaled gradient sign (FGSM attack) on the input (Kurakin et al., 2016), so that we obtain a target $SNR = 33$. Results are depicted in the left and center plots of Figure 5. In the left plot the noise is added after normalizing the input whereas on the middle plot it is added before normalizing. As in the isotropic noise case, a combination of the Parseval method and our proposed approach achieves maximum robustness.

In regards to the second approach, where a worst case scenario is considered, we use the Foolbox toolbox (Rauber et al., 2017) implementation of DeepFool (Moosavi Dezfooli et al., 2016). Due to time constraints we sample only $\frac{1}{10}$ of the test set images for this test. The conclusions are similar (right plot of Figure 5) to those obtained for the first adversarial attack approach.

### 4.3 Implementation robustness

Finally, in a third series of experiments we evaluate the robustness of the network functions to faulty implementations. As a result, approximate computations are made during the

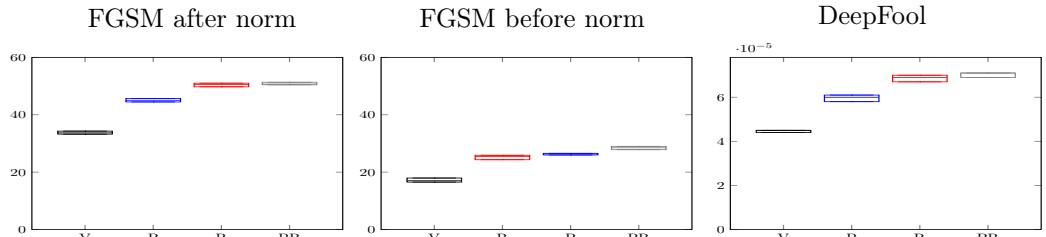

Figure 5: Robustness against an adversary measured by the test set accuracy under FGSM attack in the left and center plots and by the mean $\mathcal{L}_2$ pixel distance needed to fool the network using DeepFool on the right plot.

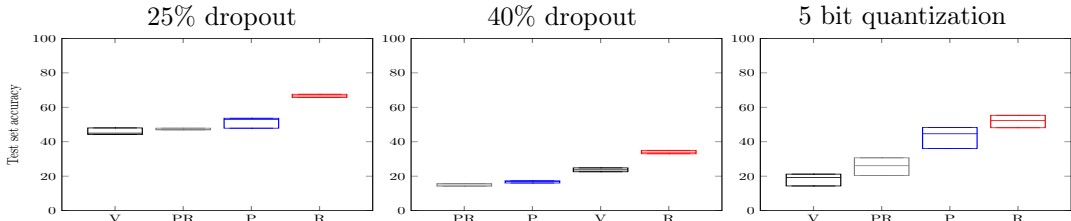

Figure 6: Test set accuracy under different types of implementation related noise.

test phase that consist of random erasures of the memory (dropout) or quantization of the weights (Hubara et al., 2017).

In the dropout case, we compute the test set accuracy when the network has a probability of either 25% or 40% of dropping a neuron's value after each block. We run each experiment 40 times. The results are depicted in the left and center plots of Figure 6. It is interesting to note that the Parseval trained functions seem to drop in performance as soon as we reach 40% probability of dropout, providing an average accuracy smaller than the vanilla networks. In contrast, the proposed method is the most robust to these perturbations.

For the quantization of the weights, we consider a scenario where the network size in memory has to be shrink 6 times. We therefore quantize the weights of the networks to 5 bits (instead of 32) and re-evaluate the test set accuracy. The right plot of Figure 6 shows that the proposed method is providing a better robustness to this kind of deformation than the tested counterparts.

## 5 Conclusion

In this paper we have introduced a new regularizer that enforces small variations of the smoothness of label signals on similarity graphs obtained at intermediate layers of a deep neural network architecture. We have empirically shown with our tests that it can lead to improved robustness in various conditions compared to existing counterparts. We also demonstrated that combining the proposed regularizer with existing methods can result in even better robustness for some conditions.

Future work includes a more systematic study of the effectiveness of the method with regards to other datasets, models and deformations. Recent works shown adversarial noise is partially transferable between models and dataset (Moosavi-Dezfooli et al., 2017; Papernot et al., 2016b) and therefore we are confident about the generality of the method in terms of models and datasets.

One possible extension of the proposed method is to use it in a fine-tuning stage, combined with different techniques already established on the literature. An extension using a combination of input barycenter and class barycenter signals instead of the class signal could be interesting as that would be comparable to (Zhang et al., 2017). In the same vein, using random signals could be beneficial for semi-supervised or unsupervised learning challenges.

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

## A   Parseval Training and implementation

We compare our results with those obtained using the method described in (Cisse et al., 2017). There are three modifications to the normal training procedure: orthogonality constraint, convolutional renormalization and convexity constraint.

For the orthogonality constraint we enforce *Parseval tightness* (Kovačević and Chebira, 2008) as a layer-wise regularizer:

$$R_\beta(W^\ell) = \frac{\beta}{2}\|W^{\ell\top}W^\ell - I\|_2^2, \tag{7}$$

where $W_\ell$ is the weight tensor at layer $\ell$. This function can be approximately optimized with gradient descent by doing the operation:

$$W^\ell \leftarrow (1+\beta)W^\ell - \beta W^\ell W^{\ell\top}W^\ell. \tag{8}$$

Given that our network is smaller we can apply the optimization to the entirety of the $W$, instead of 30% as per the original paper, this increases the strength of the Parseval tightness.

For the convolutional renormalization, each matrix $W^\ell$ is reparametrized before being applied to the convolution as $\frac{W^\ell}{\sqrt{2k_s+1}}$, where $k_s$ is the kernel size.

For our architecture the inputs from a layer come from either one or two different layers. In the case where the inputs come from only one layer, $\alpha$ the convexity constraint parameter is set to 1. When the inputs come from the sum of two layers we use $\alpha = 0.5$ as the value for both of them, which constraints our Lipschitz constant, this is softer than the convexity constraint from the original paper.

## B HYPERPARAMETERS

We train our networks using classical stochastic gradient descent with momentum (0.9), with batch size of $b = 100$ images and using a L2-norm weight decay with a coefficient of $\lambda = 0.0005$. We do a 100 epoch training. Our learning rate starts at 0.1. After half of the training (50 epochs) the learning rate decreases to 0.001.

We use the mean of the difference of smoothness between successive layers in our loss function. Therefore in our loss function we have:

$$\mathcal{L} = CategoricalCrossEntropy + \lambda WeightDecay + \gamma\Delta \tag{9}$$

where $\Delta = \frac{1}{d-1}\sum_{\ell=1}^{d}|\delta_\sigma^\ell|$. We perform experiments using various powers of the Laplacian $m = 1, 2, 3$, in which case the scaling coefficient $\gamma$ is put to the same power as the Laplacian.

We tested multiple parameters of $\beta$, the Parseval tightness parameter, $\gamma$ the weight for the smoothness difference cost and $m$ the power of the Laplacian. We found that the best values for this specific architecture, dataset and training scheme were: $\beta = 0.01, \gamma = 0.01, m = 2, k = b$.

## C DEPICTION OF THE NETWORK

Figure 7 depicts the network used on all experiments of sections 3 and 4. $f = 64$ is the filter size of the first layer of the network. Conv layers are 3x3 layers and are always preceded by batch normalization and relu (except for the first layer which receives just the input). The smoothness gaps are calculated after each ReLU.

Figure 7: Depiction of the studied network

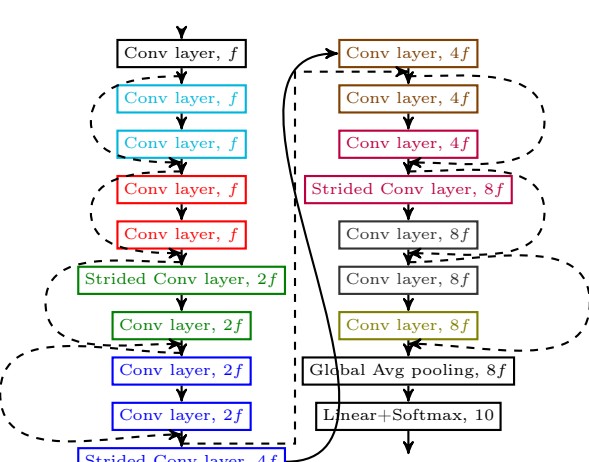

# D    ADDITIONAL EXPERIMENTS

Given suggestions from the reviewers, we performed additional experiments to further demonstrate the capabilities of the proposed regularizer. Due to the lack of space they could not be added to the main paper. We consider the effects of the regularizer when applied on another datasets. We also consider the effects of adding adversarial data augmentation methods while minimizing the amount of other influencing factors. We first look at the results when using the same architecture as for the CIFAR-10 dataset, which inevitably results in far from state-of-the-art accuracy on CIFAR-100. Then, we perform experiments using a different architecture (namely WideResnet 28-10, with dropout) for CIFAR-100.

## D.1    CIFAR-10

We add two types of tests for the CIFAR-10 dataset: adversarial data augmentation during training and black-box FGSM.

### D.1.1    TESTS WITH FGSM ADVERSARIAL DATA AUGMENTATION

In this section we consider tests adding adversarial data augmentation as suggested in (Kurakin et al., 2016). To be more precise we use the method they advise which is called "step1.1" using $\epsilon = \frac{8}{255}$. The results presented in the figures below are obtained by running 10 experiments with random initializations. We first perform the same tests as in Section 4.

As expected, we observe in Figure 8 that training with adversarial examples help in the case of Gaussian noise, as it adds more variation to the training set, while reducing the accuracy on the clean set. Note that combining our method with adversarial training results in the best median accuracy. Combining the three methods is less successful than expected, which could indicate that a better hyperparameter search would be needed.

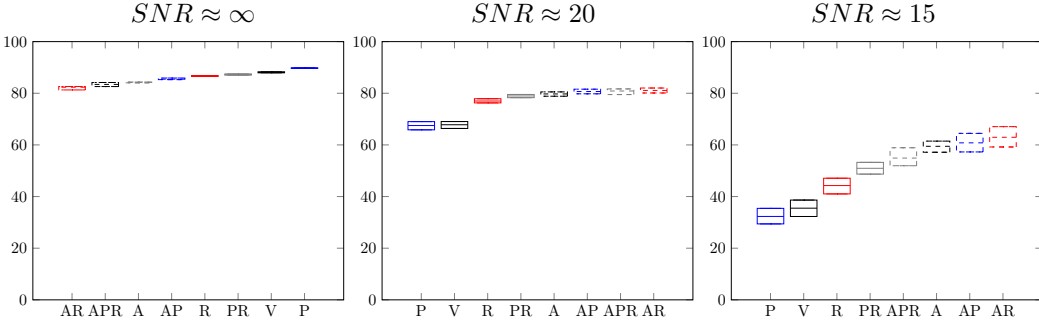

Figure 8: Test set accuracy under Gaussian noise with varying Signal-to-Noise Ratio (SNR). A is for Adversarial, P is for Parseval, R is for the proposed Regularizer and V is for Vanilla network.

Considering adversarial robustness, the obtained results are depicted in Figure 9. We observe that adding FGSM adversarial training does not generalize well to other types of attack (which is readily seen in the literature Madry et al. (2018)). Overall, the models using the proposed regularizer are the most robust.

Finally, when considering implementation related perturbations, the results depicted in Figure 10 are consistent with the ones from Section 4.3, in which is shown that the proposed regularizer helps improving robustness.

In summary, even when adding adversarial training, the proposed regularizer is either the most robust in median, or capable of improving the robustness when used combined with the other methods.

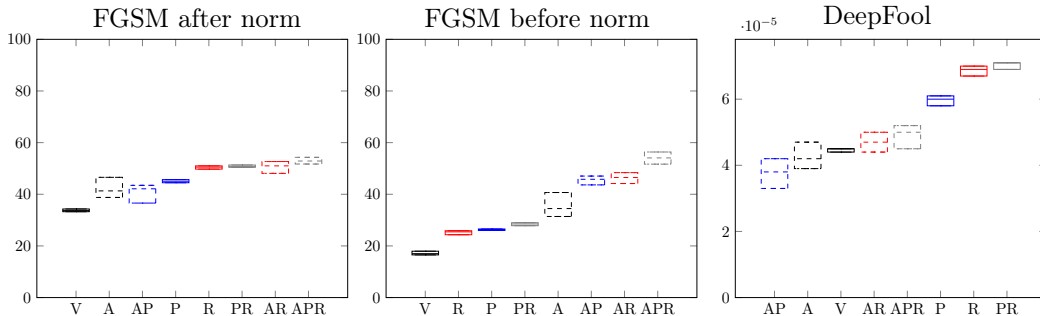

Figure 9: Robustness against an adversary measured by the test set accuracy under FGSM attack in the left and center plots and by the mean $\mathcal{L}_2$ pixel distance needed to fool the network using DeepFool on the right plot.

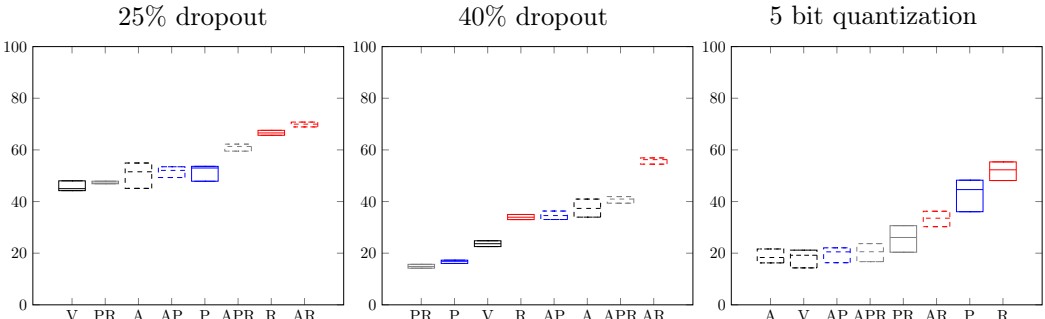

Figure 10: Test set accuracy under different types of implementation related noise.

### D.1.2  TESTS WITH BLACK BOX FGSM

To further verify that the obtained results are not only due to gradient masking, we perform tests with black box FGSM, where the target attacked network is not the same as the source of the adversarial noise.

For this test we set the SNR of FGSM to 33. We chose the network with the best performance for each of the tested methods. The results are depicted in Table 1. In our experiments, we found that the combination of our method with Parseval is the most robust to noise coming from other sources, while the noise created by both Parseval and our method did not generalize as well as the one created by Vanilla. This demonstrates that the improvements are not caused by gradient masking, but are caused by the increased robustness of the proposed method and Parseval's.

Table 1: Black box FGSM applied to the different methods. The most robust target for a given source is bolded, while the strongest source for a target is in italic.

| Target | Source | | | |
|---|---|---|---|---|
| | Vanilla | Parseval | Regularizer | Parseval + Regularizer |
| Vanilla | X | *60.74* | 61.49 | 72.51 |
| Parseval | *57.82* | X | 68.21 | **73.87** |
| Regularizer | *69.72* | 74.96 | X | 73.56 |
| Parseval + Regularizer | **75.35** | **76.11** | *70.22* | X |

### D.1.3  TESTS WITH PGD ADVERSARIAL DATA AUGMENTATION

Most of our adversarial tests are performed with FGSM because of its simplicity and speed, even though it has already been shown (e.g: Madry et al. (2018)) that FGSM is weak as an attack and as a defense mechanism. Despite the fact we do not only target adversarial

defense, we further stress the ability of the proposed regularizer to improve it and to combine with other methods. To this end we perform experiments against the PGD (Projected Gradient Descent) attack.

PGD is an iterative version of FGSM, which run for a maximum number of iterations $it$ or until convergence. For each iteration it moves by a distance of $step$ in the direction of the gradient provided it does not go at a distance greater than $\epsilon$ from the original image.

Our experiments show that the proposed regularizer increases robustness against a weak PGD attack (similar epsilon as our FGSM with SNR=33), but it is almost completely defeated by the PGD with the parameters from (Madry et al., 2018). The results are depicted in table 2. We also show that, as expected, FGSM training does not add significant robustness against the stronger PGD attack.

Table 2: Test set accuracy on the CIFAR-10 dataset against the PGD attack with different parameters.

| Model | $it = 20, step = 0.002, \epsilon = 0.01$ | $it = 20, step = \frac{2}{255}, \epsilon = \frac{8}{255}$ |
|---|---|---|
| Vanilla | 0.95% | 0.02% |
| Proposed Regularizer | 11.18% | 0.09% |
| FGSM | 5.78% | 0.09% |
| FGSM + Regularizer | **12.91%** | 0.55% |

As the proposed regularizer can be combined with FGSM defense, it is natural to also test it alongside PGD training. We use the parameters advised in (Madry et al., 2018): 7 iterations with $step = 2/255$, and $\epsilon = 8/255$. The results depicted in Table 3 show that using our regularizer increases robustness of networks trained with PGD. Note that Dropout and Gaussian Noise were applied ten times to each of the networks and the results are displayed as the mean test set accuracy under these perturbations. A rate of 40% was used for dropout. The PGD attack uses the following parameters: $it = 20, step = \frac{2}{255}, \epsilon = \frac{8}{255}$ .

Table 3: Results on the CIFAR-10 with PGD training and the hyperparameters from Appendix B.

| Robustness | Isotropic | | Adversarial | Implementation |
|---|---|---|---|---|
| Model/Test Type | $SNR \approx \infty$ | $SNR \approx 15$ | PGD | Dropout |
| PGD Training | 76.39% | 71.25% | 32.78% | 35.20% |
| PGD Training + Regularizer | 76.36% | **72.26%** | **33.72%** | **55.63%** |

## D.2  CIFAR-100

We test the generality of the method using the CIFAR-100 dataset. Results are shown in Table 4 as the mean over three different initializations. Dropout and Gaussian Noise are applied ten times to each of the networks for a total of 30 different runs. An SNR of 33 is used for FGSM, and a rate of 25% is used for dropout. Images are normalized in the same way as the experiments with CIFAR-10. Due to time constraints we sample only $\frac{1}{10}$ of the images from the test set for the Deep Fool test.

The proposed regularizer is the most robust on all categories, while Parseval has problems with the perturbations, despite yielding the best accuracy on the clean test set. The combination of the proposed regularizer and the parseval training method is not able to reproduce the good results from the CIFAR-10 dataset.

The results shown in Table 4 are obtained using an architecture that is not performing very well on the clean test set for the CIFAR-100 dataset. We thus performed additional experiments using the WideResNet 28-10 (Zagoruyko and Komodakis, 2016) architecture, and we added standard data augmentation (random crops and random horizontal flipping) and dropout with probability of 30% after the first convolution of each residual block. We train for 200 epochs, starting with a learning rate of 0.1 and divide the learning rate by 5 in epochs 60, 120 and 160. Momentum of 0.9 is used and weight decay of 5e-4. We use the

Table 4: Results on the CIFAR-100 dataset with the hyperparameters from Appendix B. Bolded value represent the best model on the test.

| Robustness | Isotropic | | Adversarial | | Implementation |
|---|---|---|---|---|---|
| Model/Test Type | $SNR \approx \infty$ | $SNR \approx 15$ | FGSM | Deep Fool | Dropout |
| Vanilla | 62.38% | 12.78% | 5.70% | 1.7E-5 | 8.66% |
| Parseval | **63.61%** | 10.11% | 5.85% | 1.5E-5 | 10.61% |
| Proposed Regularizer | 60.06% | **21.14%** | **6.15%** | **2.9E-5** | **21.40%** |
| Proposed + Parseval | 56.64% | 20.01% | 4.07% | 1.8E-5 | 9.41% |

value from the Parseval paper ($\beta = 0.0003$) as in this case it provided better results than the one described in Section B.

Results on the WideResNet 28-10 architecture using data augmentation are shown in Table 5. We observe that the proposed method (sometimes with combinations with other methods) is still the most robust.

Table 5: Results on the CIFAR-100 dataset with WideResNet 28-10.

| Robustness | Isotropic | | Adversarial | | Implementation |
|---|---|---|---|---|---|
| Model/Test Type | $SNR \approx \infty$ | $SNR \approx 15$ | FGSM | Deep Fool | Quantization |
| Vanilla | **78.42%** | 11.68% | 21.38% | 5.3E-5 | 12.56% |
| Parseval | 77.71% | 12.75% | 22.73% | 5.7E-5 | 1.58% |
| Proposed Regularizer | 77.33% | 14.46% | 23.27% | 5.8E-5 | **17.01%** |
| Proposed + Parseval | 76.72% | **20.24%** | **25.85%** | **6.9E-05** | 1.0% |

## E  IMPACT OF THE PROPOSED REGULARIZER ON THE BOUNDARY

We look at the impact of the proposed regularizer on the boundary region. To this end, we choose 10 pairs of points in distinct classes that are the most similar (i.e. their distance is minimal) in the input space and we look at the decision of the network function along the segment between them. The average is depicted in Figure 11. Note that the point to the left is always chosen to be the one corresponding to the decision of the network at the middle of the segment, so that the average curve is asymmetric.

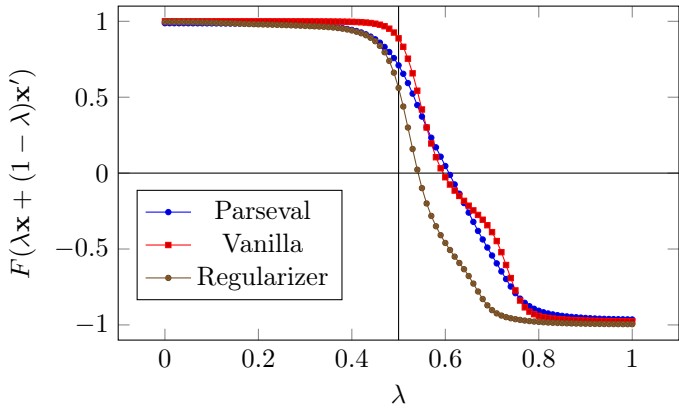

Figure 11: $F(\lambda \mathbf{x} + (1 - \lambda)\mathbf{x}')$ for different methods.

Interestingly, we observe that the proposed regularizer is the one for which the boundary is closest to the middle of the segments, thus proving our claim that the proposed regularizer control the boundary region.

## F  REGULARIZER PSEUDO-CODE

Below in Algorithm 1 we describe how we use the proposed regularizer to compute the loss as a pseudo-code. This function receives five inputs:

1. $list_{activations}$: the list of the intermediate features right after each call of the ReLU activation function of the network. We call these intermediate features **activations**$^\ell$ where $\ell$ represents the depth of the network;

2. **y**: the output of the network;

3. **s**: the label signal of the batch. Otherwise said, the ground truth labels of the examples of the batch;

4. $m$: the power of the Laplacian for which we wish to compute the smoothness;

5. $\gamma$: the scaling coefficient of the regularizer loss.

---

**Algorithm 1:** Loss function of the regularized network

---

1: **procedure** SMOOTHNESS(**activations**$^\ell$, **s**, $m$)
2:     $\mathbf{A}^\ell \leftarrow$ Pairwise cosine similarity of **activations**$^\ell$
3:     $\mathbf{D}^\ell \leftarrow$ Diagonal degree matrix of $\mathbf{A}^\ell$
4:     $\mathbf{L}^\ell \leftarrow \mathbf{D}^\ell - \mathbf{A}^\ell$
5:     $\sigma^\ell \leftarrow \text{Trace}(\mathbf{s}^\intercal (L^\ell)^m \mathbf{s})$
6:     **return** $\sigma^\ell$
7: **procedure** LOSS($list_{activations}$, **y**, **s**, $m$, $\gamma$)
8:     **for activations**$^\ell \in list_{activations}$ **do**
        $\sigma^\ell \leftarrow$ Smoothness(**activations**$^\ell$, **s**, $m$)
9:     $\Delta \leftarrow \frac{\sum_{i=1}^{\ell_{max}} |\sigma^i - \sigma^{i-1}|}{\ell_{max}-1}$
10:     **return** CategoricalCrossEntropy(**s**, **y**) $+ \gamma^m \Delta$

---

