# OpenReview forum: "Laplacian Networks: Bounding Indicator Function Smoothness for Neural Networks Robustness"
_ICLR.cc/2019/Conference_

### Official Review · AnonReviewer3 · 2018-10-29
**Concerns in its significance**

**Rating:** 5
**Confidence:** 4

**Review:**

To improve the robustness of neural networks under various conditions, this paper proposes a new regularizer defined on the graph of the training examples, which penalizes the large similarities between representations belonging to different classes, thus increase the stability of the transformations defined by each layer of the network.

The paper is overall well written, and the idea involving the Laplacian of the similarity graph is interesting. I have reviewed this paper before. Compared to the previous version, this paper made a good improvement in its experimental results, by adding two different robustness settings in section 4.1 and section 4.3, and also include DeepFool as a strong attack method for testing adversarial robustness.

However, my main concern about the paper is still about its significance.
1. It is still not clear why would this regularization help robustness especially when considering adversarial examples. Example 1 seems not obvious to me why maintaining the boundary margin (rather than expanding or shrinking) is preferred. As stated in the second paragraph in section 3.4, “lower value of \sigma^\ell(s) are indicative of better separation between classes”, what is the reason of not directly penalizing this value, rather than requesting a “stability” property on this value? How is this stability related to the robustness? This would request a deeper analysis and more empirical proofs in the paper.
2. Experimental results still seem not convincing to me. On one hand, based on the reported result, I am not very convincing that the proposed method outperforms Parseval, especially when considering the inconsistent behaviour of “Proposed + Parseval”. On the other hand, for adversarial robustness, the authors should have compared to the method of adversarial training as well. Beyond that, the authors should also be careful of the gradient masking effect of the proposed method. I am not sure if there is some other obvious benchmarks should be included for the other two robustness settings.

Other comments:
1. Descriptions in the last 3 paragraphs in section 3.2 are not very clear. It always took me a while to figure it out every time I read the paper. It would be very helpful if the computation process and the discussions can be separated here, maybe with a pseudo-code for computing the regularizer.
2. On the other hand, while the proposed regularizer can be interpreted in a perspective of the Laplacian of the similarity graph, the third part in Equation (4), that expresses the smoothness as the sum of similarities between different classes, seems more intuitive to me. Emphasizing in this interpretation may also help convey the message.

---

> ### Author Response · Authors · 2018-11-21
> **Response to reviewer 3**
>
> We would like to thank the reviewer for their comments and suggestions. We greatly appreciate that they acknowledged the paper improved since the last submission.
>
> Regularization and robustness:
> To answer the first point, we added a discussion in the supplementary material (see also answer to reviewer 1). In particular, a regularizer that aims at minimizing the quantity $\sigma^\ell$ would result in dilation in space between examples of distinct classes. The expected consequence of this would be to transform small variations in the input to large variations in the output, which is the opposite of the desired robustness behavior. Also, the boundary region would likely fall into a ``stretched'' part of the space, resulting in sharp transitions between classes. As shown in [Zhang et al 2017], this is not a desirable property as far as the generalization is concerned. We also added experiments to show the connection between the proposed regularizer and the boundary region. In Figure~11 of the supplementary material, we draw the average network function decision along segments between two input examples in distinct classes. As is shown in this figure, the proposed regularizer results in a boundary closer to the middle of the segment, thus yielding better robustness along this axis.
>
> Experimental results: To better address the significance concerns, we have multiple experiments in the supplementary material to stress the abilities of the proposed method. We tested against another dataset (CIFAR-100) and with new architecture/hyperparameters (WideResnet and standard data augmentation), and in those experiments proposed regularizer was always the most robust (sometimes in combination with Parseval). Concerning adversarial training, we have tested the proposed regularizer in combination with methods from both [Kurakin 2017,Madry 2018], and the results show that the proposed regularizer is able to increase robustness even when used in combination with adversarial training.
>
> Gradient Masking: Experiments with Gaussian noise can be seen as evidence that the robustness of our proposed method is not due only to gradient masking (since Gaussian noise is added without knowledge of the gradients). We also added black box tests, where the attacks are generated on a network trained using a distinct method from the one it is used upon, to the supplementary material. We obtained that for all sources, the most robust target network was the one that combined the proposed regularizer with Parseval, proving that the obtained results are not solely due to gradient masking, but to the increased robustness due to the use of the regularizer.
>
> Pseudo-code: In regards to pseudo-code, we will be adding one to the additional material and we are willing to share the code as soon as the review process is over, as we think that is the easiest way to explain the method and increase reproducibility. About presentation, we added a paragraph after Equation (4) to ease readability: ``In this paper we are particularly interested in smoothness of the label signals. We call \emph{label signal} $\mathbf{s}_c$ associated with class $c$ a binary ($\{0,1\}$) vector whose nonzero coordinates are the ones corresponding to input vectors of class $c$. In other words, $\mathbf{s}_c[i] = 1 \Leftrightarrow (\mathbf{x}_i$ is in class $c ), \forall 1\leq i\leq b$. Using Equation~(4), we obtain that the smoothness of the label signal $\mathbf{s}_c$ is the sum of similarities between examples in distinct classes. Thus a smoothness of 0 means that examples in distinct classes have 0 similarity.''.
>
> References:
> [Kurakin 2017] Kurakin, Alexey, Ian Goodfellow, and Samy Bengio. "Adversarial machine learning at scale." ICLR 2017.
> [Zhang 2018] Zhang, Hongyi, et al. "mixup: Beyond empirical risk minimization." ICLR 2018.
> [Madry 2018] Madry, Aleksander, et al. "Towards deep learning models resistant to adversarial attacks." ICLR 2018.

---

> > ### Comment · AnonReviewer3 · 2018-11-26
> > **Thanks for the reply.**
> >
> > It is true that minimizing the margin may be leading to some dilations. On the other hand, I cannot see how this regularizer can help in stabilizing the output neither. Quoted from the rebuttal to R1. “we would argue that by avoiding large changes in the margin (in average) from one layer to the next, the proposed regularizer aims at enforcing that small perturbations in the input cause small changes in the output.” I think the margin is more of a necessary condition rather than a sufficient condition, especially considering (1) it is an average sense; (2) the cosine similarity for the pre-layer and post-layer are two metrics in two different spaces. I also find Figure~11 may be selective, in the sense that this result actually fails to explain the outperformance of Parseval to Vanilla.
> >
> > Experimental results:
> > (1) The interaction of this regularizer with the Parseval still remains mysterious, which makes my concerns persist. Is it actually a contradict objective with the Parseval regularizer? Is it related to the model capacity?
> > (2) I don’t think the Gaussian noise can be seen as evidence for gradient masking because of its low efficiency. In general think about the efficiency difference between sampling and optimization (when it is possible).
> > (3) Overall, the results on CIFAR-10 with PGD attack in the appendix is way lower than the state-of-the-art results. Check https://github.com/MadryLab/cifar10_challenge. I think further justification on the significance of these results is necessary.

---

> > > ### Author Response · Authors · 2018-11-27
> > > **Response to reviewer 3 (1/2)**
> > >
> > > We would like to thank the reviewer again for its comments. We are sorry but due to the character limit we had to split the response in two. This is the first part:
> > >
> > > Stability of the output: Figure 11 shows that the proposed regularizer helps to preserve the margin from one layer to the next better than Parseval or the Vanilla network. We agree that this is more of a necessary than a sufficient condition, and probably that is the main reason that our model works so well when combined with Parseval, which on its own could completely collapse the margin. About the two points, we consider them as necessary relaxations in order not to prevent the network function to converge during training. Also, controlling the margin for a layer function would control its Lipschitz constant in the directions yielded by pairs of examples, provided the layer function would be linear. Layer functions are not exactly linear, but almost. In a way, considering the simplification that layer functions are almost linear, it is a stricter condition than Parseval as it enforces no dilatation nor contraction, but applied to a "small" number of directions in space, and not the whole space.

---

> > > ### Author Response · Authors · 2018-11-27
> > > **Response to reviewer 3 (2/2)**
> > >
> > > We would like to thank the reviewer again for its comments. We are sorry but due to the character limit we had to split the response in two. This is the second part:
> > >
> > > Experimental
> > > (1): We believe that our regularizer interacts well with Parseval as Parseval tries to force the Lipschitz constant to be \leq 1, which upper bounds the difference between the input and output distances, but does not provide any lower bound on it. This means that Parseval by itself does not constrain the amount of contraction. By making sure distances remain unaltered between pairs of examples during training, we avoid sharp transitions that we believe are the cause to reduced robustness of the network function. Also, it is worth pointing out that Parseval mechanism disregard the batchnorm layers (forcing the constraint on them prevent the network function from converging during training), so that the actual Lipschitz constant of the layers is not fully controlled.
> > >
> > >
> > > (2) First we would like to note that the proposed regularizer does not add operations to the network, which can easily lead to incidental gradient obfuscation, and does not modify the cross entropy landscape directly. Our regularizer is added in the same way as an L2 regularization would be added, i.e., as an extra term in the loss function that is not taken into account during the adversarial example gradient computation.
> > >
> > > Considering our experimental results:
> > > - we believe that while Gaussian Noise is not as efficient as adversarial noise, it can still be used as a first measure of robustness.
> > > - In our experiments, the proposed regularizer was robust to Dropout perturbations, which also do not use the gradient.
> > > - On the supplementary material (Table 1), there is a black box FGSM setting, where the proposed regularizer has the best performance for the examples generated from gradients of the other networks (Vanilla, Parseval).
> > > - Finally, we consider the five criteria for identifying obfuscated gradients from [Athalye 2018], and note that some of our experimental results already perform 4 of these checks, thus providing some evidence that our system should not be obfuscating gradients. In each of the following items, the condition, if met, will imply that the method obfuscates gradients, and "better" means that attacks are more successful:
> > >
> > > i) "One-step attacks perform better than iterative attacks": The results from Table 2 show that in our case iterative attacks are stronger (hence better) than one step attacks on the proposed regularizer;
> > > ii) "Black-box attacks are better than white-box attacks":  The results from Table 1 show that black box attacks are less effective than white box attacks on the proposed regularizer;
> > > iii) "Unbounded attacks do not reach 100% success": Table 2 shows that the simple PGD from [Madry 2018] fools the network designed with the proposed regularizer, therefore an unbounded attack would reach 100% success;
> > > iv) "Random sampling finds adversarial examples": This is the only one that we have not tested directly. But we believe that it should not happen, as the models also show robustness to white noise;
> > > v) "Increasing distortion bound does not increase success": The results from Table 2 show that increasing the distortion bound increases the success of the attack against the proposed regularizer.
> > >
> > > (3) Given the capacity of the network we use and the fact that it was trained only with adversarial data augmentation (and not with adversarial + standard) both of the results are far away from state of the art. Yet we believe that these results are significant as they demonstrate the effectiveness of the proposed regularizer. To better address your point, we are currently running tests using the same parameters from Madry, but in our actual settings (madry tensorflow code + our GPU) it seems that it takes a few days to obtain the results (it takes 4 seconds per step, the paper suggests 80000 steps). We will report the results back as soon as possible. Finally, we want to point out again that our main claim here is to show that we can increase the robustness to several types of perturbation, and not only adversarial ones.
> > >
> > > [Athalye 2018] Anish Athalye, Nicholas Carlini, David Wagner; "Obfuscated Gradients Give a False Sense of Security: Circumventing Defenses to Adversarial Examples"; ICML 2018

---

> > > ### Author Response · Authors · 2018-12-11
> > > **Results using code from https://github.com/MadryLab/cifar10_challenge**
> > >
> > > We are sorry for the delay, we have the results using the code from the provided github (which we will be uploading to github as soon as the ICLR review process is over).
> > >
> > > We compared the two networks provided in the madrylab github (secret and adv_trained) with 2 experiments using our regularizer with m=8 and gamma 0.01. We got improvements in the accuracy over the provided networks which we report below as the mean over the experiments:
> > >
> > > Model/Test type                  & Clean Test Set & Gaussian Noise & PGD       & CW      \\
> > > Madry 2018                          & 87.20\%           & 77.24\%              & 45.74\% & 46.63\% \\
> > > Madry 2018 + Regularizer & 87.47\%           & 78.84\%              & 45.82\% & 46.78\%
> > >
> > > PGD and CW are applied following the methodology of Madry 2018 (step_size = 2/255, epsilon = 8/255, steps = 20 and 30 respectively). Gaussian Noise is applied with SNR \approx 15 and as the mean over 10 tests for each experiment.
> > >
> > > Lastly, we want to thank again the reviewers for their valuable comments and to mention again that the objective of our work is to find better intermediate representations that are more robust to a vast number of possible deformations, not to focus on adversarial examples.

---

### Official Review · AnonReviewer1 · 2018-11-05
**interesting idea but the significance of the experimental results is unclear and the motivation should better match the evaluation**

**Rating:** 5
**Confidence:** 3

**Review:**

The paper proposes to use a regularization which preserves nearest-neighbor smoothness from layer to layer. The approach is based on controlling the extent to which examples from different classes are separated from one layer to the next, in deep neural networks. The criterion computes the smoothness of the label vectors (one-hot encodings of class labels) along the nearest-neighbor graph constructed from the euclidian distances on a given layer's activations. From an algorithmic perspective, the regularization is applied by considering distances graphs on minibatches. Experiments on CIFAR-10 show that the method improves the robustness of the neural networks to different types of perturbations (perturbations of the input, aka adversarial examples, and quantization of the network weights/dropout0.

The main contribution of the article is to apply concepts of graph regularization to the robustness of neural networks. The experimental evaluation is solid but the significance is unclear (error bars have rather large intersections), and there is a single dataset.

While the overall concept of graph regularization is appealing, the exact relationship between the proposed regularization and robustness to adversarial examples is unclear. There does not seem to be any proof that adersarial examples are supposed to be classified better by keeping the smoothness of class indicators similar from layer to layer. Section 3.4 seem to motivate the use of the smoothness from the perspective of preventing overfitting. However, I'm not sure how adversarial examples and the other forms of perturbations considered in the experiments (e.g., weight quantization) are related to overfitting.

strengths:
- practical proposal to use graph regularization for neural network regularization
- the proposal to construct graphs based on the current batch makes sense from an algorithmic point of view


cons: experimental results are a bit weak -- the most significant results seem to be obtained for "implementation robustness", but it is unclear why the proposed approach should be particularly good for this setting since the theoretical motivation is to prevent overfitting. The results vs Parseval regularization and the indications that the metohd works well with Parseval regularization is a plus, but the differences on adversarial examples are tiny.

other questions/comments:
- how much is lost by constructing subgraphs on minibatches only?
- are there experiments (e.g., on smaller datasets) that would show that the proposed method indeed regularizes and prevents overfitting as motivated in Section 3.4?

---

> ### Author Response · Authors · 2018-11-21
> **Response to reviewer 1**
>
> We would like to thank the reviewer for providing constructive and relevant comments. In the next paragraphs we address each of them.
>
> Regularization and adversarial examples: We agree that the relationship between the proposed regularizer and adversarial examples should be clarified. But first, we would like to point out that adversarial examples are considered in the paper as just one example of perturbation: our main goal is not adversarial defense, but robustness over various kinds of noise or perturbations. In this context, we would argue that by avoiding large changes in the margin (in average) from one layer to the next, the proposed regularizer aims at enforcing that small perturbations in the input cause small changes in the output. This is supported by Figure~11 in the supplementary material, where we look at the boundary region along segments between pairs of examples in distinct classes. We observe that the proposed regularizer results in a boundary region that is closer to the middle of the segment, thus providing a better robustness to deformations (at least along this axis).
>
> Overfitting: The use of the word ``overfitting'' in the introduction was misleading. We reworded it.
>
> Significance: In order to address significance issues, we ran additional experiments that are included in the supplementary material. In these experiments we added adversarial training as suggested in both [Kurakin 2017, Madry 2018] and combinations with the proposed method. We also added tests with CIFAR-100 and with another architecture (WideResNet 28-10). Interestingly, in all those experiments, the most robust method always included the proposed regularizer. In particular, despite being agnostic of the adversarial attack on which it is tested, the proposed regularizer helps even on top of strong adversarial training. We consider Table 3 to be the most significant of the new experiments.
>
> Subgraphs on mini-batches: This is an interesting question that we want to study in a future work as we believe that it would be too complex to also add to this contribution. This study would also take into consideration an approach to better sample the mini-batches for the regularizer.
>
> Figures: we have reordered the methods by the median score and changed the box plots to improve readability.
>
> References:
> [Kurakin 2017] Kurakin, Alexey, Ian Goodfellow, and Samy Bengio. "Adversarial machine learning at scale." ICLR 2017.
> [Madry 2018] Madry, Aleksander, et al. "Towards deep learning models resistant to adversarial attacks." ICLR 2018.

---

### Official Review · AnonReviewer2 · 2018-11-05
**Graph-regularized NNs**

**Rating:** 9
**Confidence:** 5

**Review:**

This paper proposes the interesting addition of a graph-based regularisers, in NNs architectures, for improving their robustness to different perturbations or noise. The regularisation enforces smoothness on a graph built on the different features at different layers of the NN system. The proposed ideas are quite interesting, and integrates nicely into NN architectures.

A few paths for improvements:

- the 'optimal' choice of the power of the Laplacian, in 3.5, is eluded
- the figures are not presented ideally, nor in a very readable form - for example, their are 90-degree rotated compared to classical presentations, and the plots are hardly readable
- the might exist a tradeoff between robustness, and performance (accuracy), that seem to be explaining the proposed results (see Fawzi - Machine Learning 2018, for example)
- in 4.2, what is a mean case of adversarial noise? Also, it would be good to see the effect of the regularizer of both the 'original' network, and on the network trained with data augmentation. It is not clear which one is considered here, but it would be interesting to study both, actually.
- the second paragraph of the conclusion (transfer of perturbations) opens interesting perspective, but the problem might not be as trivial as the authors seem to hint in the text.

Overall, very interesting and nice work, which might be better positioned (especially in terms of experiments) wrt to other recent methods that propose to improve robustness in NNs.

---

> ### Author Response · Authors · 2018-11-21
> **Response to reviewer 2**
>
> We would like to thank the reviewer for the kind review and the very relevant comments.
>
> Power of the Laplacian: The theoretical analysis that inspired our work suggest that higher powers of the Laplacian are optimal in an asymptotic sense, but this requires higher complexity, is based on synthetic data, and we have observed in testing that higher values can lead to numerical issues/lack of convergence for the network. So, we have proposed $\L^2$, which already provides improved performance, and plan to study this problem in more detail in future work.
>
> Figures: We rotated all figures so that it becomes easier to read.
>
> Reference: We would like to thank the reviewer for the reference. It is indeed a very relevant work and we included a mention in the draft: ``Such a trade-off between robustness and accuracy has already been discussed in the literature. (Fawzi et al., 2018) ''.
>
> Mean case: The mean case of adversarial noise refers to a weaker attack where the adversary would only have access to one backward pass on the network. This corresponds to the FGSM attack. The experiments in the original draft were performed without data augmentation. We have added experiments in the supplementary material which include standard (crops and flipping on CIFAR-100) and adversarial ([Kurakin 2017, Madry 2018] on CIFAR-10). We note that in these new experiments, adding the proposed regularizer always increases robustness to deformations.
>
> Conclusion: We rephrased the second paragraph of the conclusion, so that solving the problem appears less trivial now: ``Recent works shown adversarial noise is partially transferable between models and dataset (Moosavi-Dezfooli et al., 2017; Papernot et al., 2016b) and therefore we are confident about the generality of the method''.
>
> References:
> [Kurakin 2017] Kurakin, Alexey, Ian Goodfellow, and Samy Bengio. "Adversarial machine learning at scale." ICLR 2017.
> [Madry 2018] Madry, Aleksander, et al. "Towards deep learning models resistant to adversarial attacks." ICLR 2018.

---

### Public Comment · (anonymous) · 2018-11-14
**Adversarial evaluation is weak**

This paper makes some claims about how the proposed approach is adversarially robust, but does not perform a sufficient evaluation to demonstrate this fact. The authors only apply FGSM, which is known to be a weak attack (See this comment by its author https://openreview.net/forum?id=SkgVRiC9Km&noteId=rkxYnt8JpQ&noteId=rkxYnt8JpQ ), and DeepFool, which many prior papers have successfully defended against. The authors should try stronger optimization-based attacks if they wish to claim adversarial robustness (Madry et al. 2018, Kuraking et al. 2017, Carlini & Wagner 2017).

---

> ### Author Response · Authors · 2018-11-21
> **Answer to concerns about adversarial evaluation**
>
> Thank you for bringing this to our attention. First we would like to point out that adversarial attacks are only one of the possible perturbations we consider in this work, and therefore we were more interested in unit testing (cf. Goodfellow) than achieving state of the art adversarial noise defense.
>
> In any case we want to point the commenter to our meta-answer where we address this question, in summary we have added PGD training and were able to increase its robustness by combining it with our method. We believe it is quite a significant result since contrary to PGD training the proposed regularizer is completely agnostic of the adversarial attack it is tested on.

---

### Author Response · Authors · 2018-11-17
**Meta response to the comments**

We would like to thank the reviewers for their constructive and relevant comments. We have updated the draft taking the comments into account.

We would like to clarify two key points (these will be addressed in our revision as well) :

1) Adversarial attacks:

Our work is not focused specifically in improving performance under adversarial attacks. Instead, our goal is to improve the robustness of the system under various types of deformations. We view as approach as completing existing methods, including adversarial training methods. To emphasize this point, we added numerous experiments in the supplementary material. In particular, in Table 3 we show that when the proposed regularizer is combined with PGD during training the combination PGD+Proposed outperforms, in some cases significantly, using PGD alone, for isotropic noise, drop outs and adversarial attacks.

2) Proposed regularizer and system robustness:

By avoiding large changes in the margin (in average) from one layer to the next, the proposed regularizer aims at enforcing that small perturbations in the input  cause small changes in the output.

We will shortly be answering the other reviewers comments.

---

### Meta-Review · Area_Chair1 · 2018-12-13

**Confidence:** 2
**Recommendation:** Reject

**Metareview:**

The paper proposes a new graph-based regularizer to improve the robustness of deep nets. The idea is to encourage smoothness on a graph built on the features at different layers. Experiments on CIFAR-10 show that the method provides robustness over very different types of perturbations such as adversarial examples or quantization. The reviewers raised concerns around the significance of the results, the reliance on a single dataset and the unexplained link between adversarial examples and the regularization. Despite the revision, the reviewers maintain their concerns. For this reason this work is not ready for publication.